# Genotype and Phenotype Landscape of 283 Japanese Patients with Tuberous Sclerosis Complex

**DOI:** 10.3390/ijms231911175

**Published:** 2022-09-22

**Authors:** Sumihito Togi, Hiroki Ura, Hisayo Hatanaka, Yo Niida

**Affiliations:** 1Center for Clinical Genomics, Kanazawa Medical University Hospital, Uchinada 920-0293, Japan; 2Department of Advanced Medicine, Division of Genomic Medicine, Medical Research Institute, Kanazawa Medical University, Uchinada 920-0293, Japan

**Keywords:** tuberous sclerosis complex, genotype–phenotype correlations, mutation detection methods, mosaic mutations, deep intronic mutations

## Abstract

Tuberous sclerosis complex (TSC) is an autosomal dominant disorder characterized by multiple dysplastic organ lesions and neuropsychiatric symptoms, caused by loss of function mutations in either *TSC1* or *TSC2*. Genotype and phenotype analyses are conducted worldwide, but there have been few large-scale studies on Japanese patients, and there are still many unclear points. This study analyzed 283 Japanese patients with TSC (225 definite, 53 possible, and 5 genetic diagnoses). A total of 200 mutations (64 *TSC1*, 136 *TSC2*) were identified, of which 17 were mosaic mutations, 11 were large intragenic deletions, and four were splicing abnormalities due to deep intronic mutations. Several lesions and symptoms differed in prevalence and severity between TSC1 and TSC2 patients and were generally more severe in TSC2 patients. Moreover, *TSC2* missense and in-frame mutations may attenuate skin and renal symptoms compared to other *TSC2* mutations. Genetic testing revealed that approximately 20% of parents of a proband had mild TSC, which could have been missed. The patient demographics presented in this study revealed a high frequency of TSC1 patients and a low prevalence of epilepsy compared to global statistics. More patients with mild neuropsychiatric phenotypes were diagnosed in Japan, seemingly due to a higher utilization of brain imaging, and suggesting the possibility that a significant amount of mild TSC patients may not be correctly diagnosed worldwide.

## 1. Introduction

Tuberous sclerosis complex (TSC) is an autosomal dominant inherited genetic disorder characterized by multiple organ lesions, epilepsy, and neuropsychiatric manifestations. Its estimated frequency in the population is about one in 6000 to 10,000, of whom approximately two-thirds are sporadic, with the remainder segregated in families [1]. TSC lesions are composed of disarranged undifferentiated dysplastic cells. In some lesions, these dysplastic cells do not grow more rapidly than normal cells (hamartia); but in other lesions, they grow into benign tumors (hamartomas) or rarely grow into malignant tumors, such as renal cell carcinoma [2]. The representative hamartia of TSC is brain cortical tubers, whose disease name is derived from macroscopic pathological findings, described by Bourneville as “tuberous sclerosis of the cerebral convolutions” [3]. In addition, representative hamartomas are facial angiofibromas (AF) and renal angiomyolipomas (AML).

In practice, entire body organs are involved in patients with TSC, including the brain, skin, retina, lung, liver, kidney, heart, pancreas, intestines, endocrine system, bone, and teeth [4], and the established clinical diagnostic criteria are widely used [1]. All hamartias are embryonic lesions, whereas hamartomas appear in an age-dependent fashion. TSC is also associated with epilepsy and various neuropsychiatric symptoms, called tuberous sclerosis-associated neuropsychiatric disorders (TAND) [5]. TAND includes many psychiatric disorders and symptoms, such as intellectual disability, learning disorders, autism spectrum disorder, attention deficit hyperactivity disorder, higher brain dysfunction, mood disorders, and so on. Despite the diverse lesions and symptoms that can occur with TSC, their combination and severity in individual patients vary widely between families or even within a family. Some patients are diagnosed early in life, because of developmental delay and epilepsy, while others are diagnosed in adulthood with renal AML as the chief complaint and without any neuropsychiatric symptoms. This diversity makes it challenging to understand the overall picture of TSC.

The disease is caused by loss of function mutations of one of two genes, *TSC1* and *TSC2*, which encode hamartin and tuberin, respectively [6,7]. These proteins form a functional heterodimer. This intracellular complex (TSC complex) is a universal sensor of several intracellular and extracellular signals, e.g., growth factor stimulation, hypoxia, and energy levels [8]. The most remarkable function of the TSC complex is the GTPase activating protein (GAP) activity for Ras homolog enriched in the brain (Rheb), which converts Rheb-GTP to Rheb-GDP. While Rheb-GTP activates the mechanistic target of rapamycin kinase (MTOR), especially in the mTORC1 complex. In cells lacking proper functioning of the TSC complex, there are increased levels of Rheb-GTP, which leads to activation of mTORC1, causing increased cell size and growth through phosphorylation of eukaryotic translation initiation factor 4E-binding protein 1 (4E-BP1), ribosomal protein S6 kinase beta-1 (S6K1), and eukaryotic translation initiation factor 2 (eEF-2K) [8,9,10].

*TSC1* and *TSC2* gene mutation patterns are also diverse, with no mutational hotspots in relatively large-size genes. Moreover, adding to point mutations, large deletions, splicing abnormalities due to deep intronic mutations, and mosaic mutations for all these mutations are included in a non-negligible proportion. This makes the genetic diagnosis of TSC difficult. To date, several large-scale TSC mutation studies have been completed worldwide. Several aspects of the molecular genetics of TSC have been historically elucidated through these studies. The overall mutation distribution among patients with TSC shows a higher frequency of *TSC2* than *TSC1* [11]. Pathogenic missense mutations in *TSC1* are rare [12]. *TSC2* mutations produce a more severe phenotype than *TSC1* mutations, particularly in intercultural disability [12,13]. Some *TSC2* missense mutations (i.e., p.Arg905Gln and p.Gln1503Pro); however, not all missense mutations are associated with milder disease phenotypes [14,15], and *TSC2*/*PKD1* contiguous gene deletion is associated with polycystic kidney disease [16].

After that, TSC research progressed further, with the launch of the tuberous sclerosis registry to increase disease awareness (TOSCA) global patient registry, which published baseline data of 2093 patients in 2017 [17]. In Japan, we analyzed 57 patients with TSC in 2013 [18]. In this study, the frequency of patients with a *TSC1* mutation was higher than that reported in previous studies, suggesting that differences in the prevalence of diagnostic brain imaging, due to differences in medical insurance systems, may result in the diagnosis of more patient populations with mild TSC. However, a more detailed analysis was impossible, due to an insufficient number of patients. Since then, we have continued to perform genetic tests of TSC for clinical purposes, receiving patient specimens from hospitals all over Japan. We have also developed a new analysis method that can adapt to various types of mutations using a single platform of next generation sequencing (NGS), named combined long amplicon sequencing (CoLAS) [19,20]. In this study, we analyzed the phenotype and genotype details of 283 TSC patients from 260 families, to clarify the overall picture of TSC in Japan.

## 2. Results

The complete genotype and phenotype dataset of the patients in this study is assembled in Appendix A. A total of 283 patients with TSC (225 definite and 53 possible clinical diagnosed cases, and five genetic diagnosed cases) were included in this study. Of these, 53 patients had previously been reported in our 2013 study [18].

### 2.1. Patient Demographics

Demographics of the participants are summarized in Table 1, and the average, median, and range of age were 16.4, 7, and 0–80 years, respectively. The male-to-female ratio was almost equal. The number of patients by age group is shown in Figure 1. These age groups were set so that the number of patients in each group was approximately equal for statistically comparing symptom prevalence between ages. Compared to the TOSCA baseline study, more patients under five years, especially infants, were enrolled. However, the overall proportions of patients under 18 and over 18 were similar.

### 2.2. Frequency of Each Lesion and Symptom

#### 2.2.1. Distribution of Each Lesion and Symptom in All Aged Patients

The frequency of each lesion and symptom was tabulated in 225 patients with clinically definite TSC. As this study includes possible TSC with undetermined disease causative mutations (*n* = 32) and patients with genetic diagnosis whose symptoms were not well evaluated, the following phenotype assessment and comparison with other studies were limited to definite TSC. The results in descending order of frequency are shown in Figure 2. Subependymal nodules (SEN) and cortical tubers were the most frequently appearing lesions, followed by hypomelanotic macules (HM) and epilepsy in TSC1 and TSC2 patients. Developmental delay/intellectual disability (DD/ID) was detected in 41.8% of patients. Notably, retinal hamartoma was mostly detected in TSC2 patients, and multifocal micronodular pneumocyte hyperplasia (MMPH) was mostly detected in TSC1 patients. Moreover, cardiac rhabdomyomas were more common in TSC2, and ungual fibroma (UF) was more common in TSC1 patients. However, it was necessary to consider the age-dependent appearance of TSC lesions.

A high testing rate was demonstrated for most lesions, but this was slightly low for retinal hamartoma and lymphangiomyomatosis (LAM), and only 31.0% for MMPH.

#### 2.2.2. Distribution of Each Lesion and Symptom in Each Age Range

Since several TSC lesions appear age-dependently, Figure 3 shows changes in the prevalence of each lesion in each age group. SEN and cortical tubers, which are embryonic lesions, were evenly distributed across all age groups, and the distributions of epilepsy and DD/ID were correspondingly uniform across age groups. The rate of severe lesions also did not differ significantly between ages, and age dependence was not observed in these lesions and symptoms. In contrast, the prevalence and severity of skin and renal lesions increased with age. The onset and peak age differed depending on the lesion. Regarding visceral lesions, renal AML appears rapidly after age 10 and peaks after age 30. Liver AML, LAM, and MMPH appear later than renal AML. Renal cysts are observed from the neonatal period, but increase after age 30. On the other hand, the frequency of cardiac rhabdomyomas peaks in infancy, and the rate of increase with age is low. Hypomelanotic macules (HM) show little variation in frequency across ages in the skin lesions. Facial angiofibroma (AF) appear before the age of 10 years and peak before the age of 30 years, and other skin lesions also tend to increase in frequency with aging. UF does not appear in children under 10 years of age. Shagreen patch (SP) and forehead plaque (FP) can occur in early childhood, but less frequently.

#### 2.2.3. Comparing to TOSCA Baseline Data

When the prevalence of symptoms and lesions were compared between this study and the TOSCA baseline data, the general distributions appeared to be consistent (Figure 1, upper panel). However, statistical analysis showed significant differences in some lesions and symptoms (Table 2). As a whole, epilepsy, DD/ID, cortical tuber, SEGA, AF, SP, and renal AML were significantly less frequent, and cardiac rhabdomyoma was significantly more frequent in this study. Since the prevalence of each lesion differs depending on age and causative gene (*TSC1* or *TSC2*), when the patients were divided into those aged 10 and over, TSC2 patients, and TSC1 patients; there was no significant difference in prevalence in at least one category for all symptoms and lesions, other than epilepsy.

#### 2.2.4. Comparing to Other Japanese TSC Cohort Data

Another large-scale study of symptom prevalence in Japanese patients with TSC was performed in 2013 by Wataya-Kaneda et al. [21]. This study included a total of 166 outpatients treated in the Department of Dermatology of Osaka University Hospital (Osaka, Japan). The average age of the patients was 26.6 years old, which is higher than that of TOSCA and present study, and because the study was conducted on outpatients of a dermatology department, the prevalence of skin lesions tended to be high. In addition, the prevalence of renal AML and LAM was high, due to the older age group. Nonetheless, the positive rates for epilepsy, DD/ID, HM, cardiac rhabdomyomas, and renal cysts were mostly consistent with the present study.

#### 2.2.5. Gender Differences in the Prevalence and Severity of Each Symptom

Differences of sex are summarized in Appendix A. All patients who developed LAM in this study were female. Among other symptoms, the overall incidence of DD/ID, and severity of DD/ID in patients aged ≥10 years were significantly higher in males. The frequency and severity of renal AML were significantly higher in females, supporting the results of the TOSCA renal AML study [22]. However, the female patients were older than the male patients in this study, and these results need to be considered with caution. In fact, when limited to those aged ≥10 years, the significant difference between sex disappeared. The higher prevalence in women could be due to a higher percentage of younger male patients before they develop AML, and further investigation is needed to draw a conclusion. A similar logic should be applied to the higher rate of severe renal cyst and the lower frequency of cardiac rhabdomyomas in female patients.

### 2.3. TSC1 and TSC2 Mutation Distributions

Various types of *TSC1* and *TSC2* mutations were detected in 174/225 (77.3%) in definite TSC, 21/53 (39.6%) in possible TSC, and 4/5 in not-evaluated-well parents of probands (one case had a *TSC2* variant of uncertain significance) (Table 3); with, as a whole, 199/283 (70.3%) of all enrolled patients and 178/260 (68.5%) probands (Table 4). The major types of mutations are those that prevent protein synthesis, and missense mutations and in-frame mutations together account for approximately 25% of all mutations. The distribution of point mutations is summarized in Figure 4. While protein truncated mutations are diffusely distributed in *TSC1* and *TSC2*, missense and in-frame mutations are concentrated in the functional domains. Missense mutations in *TSC1* were rare, and only two were detected in the Rho-activating domain. On the other hand, missense mutations in *TSC2* were mainly observed in the tuberin domain and C-terminal RAP-GAP domain. In a definite TSC patient (family number 251), both *TSC1* and *TSC2* mutations were detected. NM_000368.5(TSC1): c.363 + 1G > T promotes exon 5 skip. However, exon 5 skip is an in-frame of 153 base pairs and is also observed at low frequency in normal controls. NM_000548.5(TSC2): c.5210C>T p. (Pro1737Leu) is a previously unreported missense variant located in the RAP-GAP domain, and in-silico analysis supports a deleterious effect. Both variants are likely to be pathogenic, according to the American college of medical genetics and genomics (ACMG) guidelines [23]. Moreover, deep intronic mutations caused several splicing aberrations, and large intragenic deletions and *TSC2*/*PKD1* contiguous deletion were also observed.

This study’s proportion of TSC1 patients was significantly higher than that of the TOSCA baseline data and other previous large scale mutation studies (Table 5). Since TSC1 patients are prone to a mild phenotype, the TSC1 proportion in familial cases is higher than sporadic cases. However, even when limited to sporadic cases, the proportion of TSC1 patients was still higher than in other studies, indicating that the ratio of familial cases was not the reason. In addition, Table 5 only compares the definite TSC in this study to other studies, this is not because it includes much possible TSC with a milder phenotype.

### 2.4. TSC1 vs. TSC2 Patients (Genotype-Phenotype Correlations)

The prevalence of each symptom and lesion in TSC1 and TSC2 patients was compared among patients with confirmed gene mutations. Patients with *TSC2*/*PKD1* contiguous deletion, not-evaluated-well, and a patient with both *TSC1* and *TSC2* mutations were excluded (Table 6). Since TSC phenotypes are likely to vary even within the same family (with the same mutation), the statistical analysis included all evaluated patients in the family, in order to determine genotype–phenotype correlations more accurately. The prevalence of DD/ID, renal AML, retinal hamartoma, and cardiac rhabdomyoma was significantly higher in TSC2 than in TSC1 patients. Conversely, the prevalence of UF and MMPH was significantly higher in TSC1 patients. This significant difference did not change, even when limited to definite TSC. Owing to the age-dependent expression of TSC lesions, statistical analyses restricted to ages ≥10 and ≥20 years were performed. No significant difference in DD/ID was observed over 20 years of age. The highest significant difference was in renal AML in those over 10 years of age. Conversely, no significant difference in cardiac rhabdomyoma was observed in those over 10 years of age. A TSC1 dominance of UF prevalence was still detected in those aged ≥10 years and around the threshold level at 20 years; however, MMPH became non-significant at ≥10 years and older. No difference in the prevalence of angiofibroma was observed in different age analyses. The prevalence of epilepsy was slightly higher in TSC2 patients, but this was not statistically significant.

Next, we examined the difference in severity between TSC1 and TSC2 patients by their high score lesions (Table 7). The scoring system is shown in Table 13 in Section 4. The most severe DD/ID (score 3, DQ/IQ < 34) was rare in TSC1 patients and was significantly more frequent in TSC2 patients. Regarding facial angiofibroma (AF), the frequency of scores of 3 tended to be higher in TSC2 patients, and a statistically significant difference was obtained in those ≥10 years of age. Patients with a high number of cortical tubers (score 3, more than 6 lesions) were significantly more frequent in TSC2 patients, despite no difference in the number of SEN. The prevalence of patients with renal AML scores of 2 or 3 was significantly higher in TSC2 patients. The proportion of patients with a score of 3 (having >4 cm AML) who require treatment was significantly higher in TSC2 patients aged ≥10 years. The proportion of patients with incompletely controlled epilepsy (score 3, having active seizures) was higher in TSC2, and the difference was significant in age ≥10 years.

### 2.5. TSC2 Protein Productive vs. Not Productive Mutations

Among TSC2 patients, we examined the difference in symptoms between mutations that were expected to produce a protein (PP), including missense, in-frame, and stop codon mutations and the mutations predicted to produce truncated or no protein (NP), including frameshift, nonsense, splicing, and large deletion (Table 8). The prevalence of SEN, AF, and UF was significantly higher in NP than in PP. In addition, in AF and renal AML, the proportion of patients with severe lesions was significantly higher in NP patients.

### 2.6. Mosaic Mutations

Several mosaic mutations were detected in both *TSC1* and *TSC2* genes. Among the 179 *TSC* mutations detected in the proband, 14 (7.8%) were mosaic mutations, and parental analysis revealed three additional mosaic mutations. Overall, 17/200 (8.5%) of the detected mutations were mosaics (Table 9). There were three ways in which mosaic mutations were identified. The first was when a patient’s blood sample analysis showed mutations with a variant allele frequency (VAF) indisputably below 0.5. The second was when the mutation was detected in a proband (child), then single-site analysis was performed on both parents, and the same mutation was seen as a low-frequency mosaic in one parent. The last was that, after the mutation had been detected by analyzing the patient’s tumor lesions, the patient’s blood sample was examined for the same mutation. The second and last cases were single-site tests for known mutations, but the first case was screening mosaic mutations itself, and it was difficult to detect very low-frequency mosaicism.

### 2.7. Parental Mutation Analysis

In 49 families, a detailed genetic analysis of the parents was conducted after the mutation was identified in the proband. The results are summarized in Table 10. Of these, mutations in both parents, including low-frequency mosaicism, were not detected in 36 families, suggesting that the patients had de novo mutations. In 13 families, one of the parents had the same mutation as the proband, three of whom had mosaic mutations (Table 9 and Table 10). Notably, 10 of these parents had not been evaluated before the proband diagnosis and had not been suspected of having TSC. Six of the ten underwent a surveillance test and were clinically diagnosed as two with definite TSC and four with possible TSC. These four individuals with possible TSC would already have met genetic diagnosis criteria. These data suggest that gene mutations could be identified in about 20% of parents who do not appear to have TSC.

## 3. Discussion

### 3.1. Patient Population

This study obtained clinical symptoms from patients who underwent genetic testing. Notably, there was a selection bias because of the recruited patients who required testing. Compared to the TOSCA baseline data, a higher proportion of infants and toddlers were included in this study, as patients aged 5 years or younger accounted for 46.3% vs. 26.7% for TOSCA (Figure 1). The need for genetic testing in infants and toddler lies in estimating developmental prognosis and determining heritability when parents wish to have a second child. Genetic testing is also indicated for a neonate with cardiac rhabdomyomas, to provide the necessary treatment, as the diagnosis cannot be confirmed clinically because of no other TSC lesions. This study included six patients with such a situation, and gene mutations were detected in five of them, one *TSC1* and four *TSC2*. Cardiac rhabdomyomas were found in at least 50% of TSC and grew during late pregnancy, influenced by maternal hormones, but some showed spontaneous regression after birth. In recent years, the number of diagnoses in the fetal period has increased along with technological advances in fetal ultrasound [30]. In most cases, it is asymptomatic and does not require treatment, but it is a lesion that can cause perinatal to neonatal death and is one of the emergency states of TSC [31]. The treatment with mTOR inhibitors (everolimus, sirolimus) is effective for cardiac rhabdomyomas of TSC [32,33], and the Japanese Society of Tuberous Sclerosis has published an “Expert opinion consensus for everolimus treatment” (http://jstsc.kenkyuukai.jp/information/information_detail.asp?id=98767 (in Japanese), last accessed on 27 August 2022).

On the other hand, adult patients in this study include those whose diagnosis could not be confirmed clinically, who wished to have children, and obtained positive results from genetic testing after mutations were identified in their children. Despite these biases, the proportions of patients aged 18 years and younger and older than 18 years were similar to those in TOSCA, indicating that this study included patients of a wide age range and reflected the actual distribution of TSC patients in Japan. In Japan, the genetic testing for TSC was covered by national insurance from April 2022, and a clinical laboratory company started accepting samples. Prior to that, facilities that could perform genetic testing were limited, and our facility had more than 90% of the domestic share. This is thought to be the reason why we were able to recruit patients from a wide range of populations.

### 3.2. Prevalence of Lesions and Symptoms in Patients with TSC

The prevalence of each lesion and symptom in definite TSC patients in this study was roughly similar to that in the TOSCA baseline data (Figure 2, upper panel). However, the age distribution of patients and the ratio of *TSC1* and *TSC2* were different from that of the TOSCA, the differences of prevalence in some lesions were statistically significant in the overall comparison. A low frequency of AF, SP, and renal AML, and a high frequency of cardiac rhabdomyomas were not observed in patients older than 10 years. A low frequency of DD/ID and cortical tuber disappeared in patients with *TSC2* mutations, and a low frequency of SEGA disappeared in patients with both *TSC1* and *TSC2* mutations. These results reflected the higher percentage of young and TSC1 patients in our study population. On the other hand, the prevalence of epilepsy was consistently low, even after adjusting for age and genes, suggesting that the prevalence of epilepsy in this study is low for different reasons. In the TOSCA epilepsy study, 83.6% (1852/2216) of patients had epilepsy, of which 38.9% presented with infantile spasm (mean age at diagnosis was 0.4 year) and 67.5% with focal seizures (2.7 year) [34]. This suggests that in the TOSCA study, there are many young patients diagnosed with TSC using epilepsy as the initial symptom. On the other hand, in a previous study by Wataya-Kaneda et al. [21], which mainly targeted adult patients visiting a dermatologist, the prevalence of epilepsy was 63%, which is almost consistent with the present study. Compared with TOSCA, it seems that more patients were suspected to have TSC from symptoms other than epilepsy in this study, such as cardiac rhabdomyomas and HM in infants, AF in childhood, and renal AML in adulthood. While many patients were diagnosed as clinically definite TSC using brain imaging. In fact, in this study, brain imaging was performed in 100 (87.7%) of 114 patients without epilepsy, more than 95% chose MRI for the imaging test modality, and cortical tuber and/or SEN was detected in 56 patients (Appendix A). Indeed, it is known that the adoption rate of CT/MR units in clinics and hospitals in Japan is the world’s best [35].

High surveillance testing rates were shown for most lesions, but retinal hamartoma was slightly lower. Regarding pulmonary involvement, LAM had a low testing rate, and MMPH was the most underreported, probably due to the low disease awareness.

The appearance of lesions in TSC was age-dependent, but lesions of the central nervous system (cortical tuber, SEN, SEGA, retinal hamartoma) and HM were fetal lesions, which were consistently expressed after birth and did not change their frequency with age. As cardiac rhabdomyomas can regress spontaneously, the incidence decreased with age and leveled off after 10 years of age. Other skin and visceral lesions increased in frequency with age. Among these, AF and SP rose sharply after the age of 4. Renal AML and UF then rose after 10 years of age. Liver AML, LAM, and MMPH were even later, rising after 20 years of age, and the occurrence was low. Renal cysts and FP were observed in the neonatal period, and although they remained constant in childhood, the frequency increased from adolescence to adulthood. Moreover, the proportion of patients with severe lesions also increased in lesions that increased in frequency with age (Figure 3). These results generally agreed with past reports [4]. However, in this study, the timing of increases in AF and renal AML was not the same. The timing of increases in AF and SP was almost the same, and the increase in renal AML occurred later, similarly to UF rather than AF.

Lesions that increase with age are more likely to develop loss of heterozygosity (LOH). LOH is most frequent in renal AML and less common in central nervous system lesions [36]. Additionally, some LAMs have been shown to occur as metastatic lesions of the renal AML [37]. Such differences in onset mechanisms, haploinsufficiency, LOH, and metastasis may determine the timing of onset of each lesion. As with LAM, the possibility of metastatic lesions from renal AML should be considered for liver AML. In fact, of the 17 patients with liver AML, 15 had renal AML, and eight had a score of 3 for severe renal AML (Appendix A). Although there are few reports on MMPH, we previously reported that LOH occurs in lesions [38].

### 3.3. TSC Mutation Spectrum

Both *TSC1* and *TSC2* showed various gene mutation patterns. Characteristically, mutations predicted to have protein production were few in *TSC1,* but account for about one-third in *TSC2* (Table 4). *TSC2* missense and in-frame mutations were concentrated in two functional domains; the tuberin domain and the RAP-GAP domain (Figure 4). The tuberin domain comprises alpha helices connected by loops, in an arrangement similar to the alpha–alpha superhelix domain. The tuberin domain is thought to stabilize the structure of the tuberin protein by forming a salt bridge between a positively charged amino acid residue on one alpha helix and a negatively charged amino acid residue on another alpha helix [39]. The TSC complex acts as a GTPase activating protein (GAP) toward a small G-protein GTPase Ras homolog expressed in the brain (RHEB). The RAP-GAP domain of tuberin promotes this reaction and inhibits the RHEB–GTP-dependent stimulation of the mammalian target of rapamycin (mTOR) complex 1 (mTORC1) [40,41]. The broad sense of the RAP-GAP domain (amino acids residue 1531–1758) overlaps the core RHEB-GAP domain (1517–1674) and includes a calmodulin (CaM) binding domain (1740–1755) (Figure 4). All C-terminal *TSC2* missense mutations are contained in this region.

Compared with TOSCA baseline data and previous studies, this study’s proportion of TSC1 patients was significantly higher (Table 5). This trend was also noted in our previous report [18], and this study confirmed that more TSC1 patients were diagnosed in Japan. It is unlikely that the regional differences in the frequency of TSC1 patients mean that the barriers to brain imaging diagnosis were low in Japan for national medical insurance, so it is likely that TSC patients with milder phenotypes were more frequently diagnosed. Conversely, there might be more undiagnosed TSC1 patients worldwide. Moreover, the prevalence of epilepsy in TSC2 patients was lower than in the TOSCA study, which indicated that TSC2 patients with milder central nervous system symptoms were diagnosed more frequently in Japan.

### 3.4. Genotype-Phenotype Correlations

The prevalence of DD/ID was significantly lower in TSC1 than in TSC2 patients (Table 6). However, this significant difference disappeared in patients aged 20 years or older. In this age group, the prevalence of DD/ID in patients with TSC1 was increased, and the prevalence of patients with TSC2 was decreased. The TSC1 patients with severe neuropsychiatric symptoms continued their treatment, while TSC2 patients without neuropsychiatric symptoms with renal AML as the major chief complaint were increased. In addition, there are significant differences in the prevalence of some lesions between TSC1 and TSC2 patients. Retinal hamartoma was more common in TSC2 patients, and only two of the 28 cases were TSC1 patients. Conversely, more MMPH was found in TSC1 patients, and only one of the seven was a TSC2 patient [42]. Cardiac rhabdomyomas were also found in TSC1 patients, but were approximately twice as common in TSC2 patients.

UF was more frequent in TSC1 patients and more than three-times as common as in TSC2 patients. This significant difference was maintained even in the group of 10 years and older, and the p-value was at the threshold (0.0579) in the 20 years and older. Likely, the TSC1 significance in UF was not due to the age difference between the groups, which could be a subject for future investigation. Although the severity of UF was not assessed in this study, a previous report showed severe squamous cell carcinoma-like UF in patients with a *TSC1* frameshift mutation [43].

Comparing the severity of symptoms (Table 7), DD/ID and renal AML were significantly more severe in TSC2. Regarding epilepsy, no difference in the prevalence was observed between TSC1 and TSC2 patients, but the percentage of patients with residual seizures was significantly lower in TSC1 patients aged ≥10 years. These results suggested that epilepsy in TSC1 patients was more likely to go into remission. Similarly, there was no difference in the prevalence of AF overall. Nevertheless, the proportion of patients with severe AF lesions (score 3 in Table 13) was significantly higher in TSC2 patients aged ≥10 years. Comparing the age group ≥10 years, score 3 DD/ID and renal AML were about ten times higher in TSC2 patients than in TSC1 patients and more than five-times higher in AF (Table 7).

Previous studies have not shown a clear correlation between the type of *TSC2* mutation and disease severity. In this study, comparing *TSC2* mutations expected to produce a protein (PP) and mutations predicted to produce truncated or no protein (NP), AF and renal AML were significantly more severe in patients with NP mutations. There were no significant differences in epilepsy and DD/ID, but there were significantly more patients with NP who had six or more SENs. A comparison of TSC1 and TSC2 patients showed a higher number of cortical tubers in TSC2 patients. In contrast, a comparison of *TSC2* PP and NP showed no significant differences of cortical tubers. It is suggested that the PP mutation of *TSC2* might be a factor that predicts the milder phenotypic consequences in AF and renal AML, but not neuropsychiatric symptoms, and further analysis using more patients will be necessary in the future.

Four patients with *TSC2*/*PKD1* contiguous gene deletion syndrome were detected in this study. Of these, only one patient, 42 years old, presented with classic polycystic kidney disease, and it should be noted that young patients do not always develop polycystic kidney disease. None of the four cases were accompanied by DD/ID, and two were a mother and her daughter, indicating that contiguous *TSC2*/*PKD1* deletion does not necessarily lead to severe neuropsychiatric symptoms. In the literature, there are reports of familial cases with mild neuropsychiatric symptoms [44,45] and cases with severe mental retardation [46,47]. Conversely, point mutations of *TSC2* were detected in four out of eight patients with classic polycystic kidney disease (Appendix A). This differentiation is important, because *TSC2*/*PKD1* deletion carries the risk of polycystic kidney disease 1, PKD1 (OMIM# 173,900), associated with symptoms other than polycystic kidney disease, namely cerebral aneurysms and valvular heart disease; lesions not associated with TSC.

One patient had both *TSC1* and *TSC2* mutations (family #251), although the severity of this patient could not be said to be higher than other patients. Mutations in both genes were also detected in five patients in the TOSCA study, suggesting that a small number of such patients are likely to exist in reality. Future investigations are required, to determine whether the symptoms of such patients will become severe or remain unchanged. On the other hand, neither our study nor others reported patients with *TSC1* or *TSC2* germline mutations in both alleles simultaneously. When one cell has TSC mutations in both alleles, it is thought that it will become tumorigenic, and such a zygote is envisioned to be embryonically lethal.

### 3.5. Mosaic Mutations

Of the 200 mutations detected in this study, 17 (8.5%) were mosaic mutations (Table 9). As previously described, this was detected during mutation screening in the proband. In this case, mutations with variant allele frequencies (VAF) of 10% or more were mainly detected. However, in some cases, mosaic mutations with a lower frequency can also be seen (Figure 5). Estimated VAF is based on NGS variant callers, but in the case of large deletions, VAF could not be estimated, due to the amplification bias of PCR. Moreover, parents were tested and found to have mosaic mutations after the mutation was identified in a proband. In this situation, a single-site test was used, and it was possible to analyze lower-frequency mosaic mutations. Similarly, the mutation was detected in a patient-derived tumor specimen and whether the same mutation was also present in the blood was investigated. Four cases, three with SEGA (Family# SS, 191 and 231) and one with subcutaneous angiofibroma (Family #258), were included in this study.

Regarding SEGA, differences between sporadic SEGA and TSC have been discussed. However, patients with somatic *TSC2* mutations limited to tumors and those with the low-frequency mosaic mutation also detected in blood were reported [28,29]. Ultimately, these patients can be interpreted as a spectrum, depending on when the mosaic mutations occur during fetal development. The same may be true for sporadic LAM, which is usually considered to have no mutations other than renal AML and LAM. That is a topic for future investigation.

It is essential to consider the detection limit when analyzing mosaic mutations. Very low-frequency mosaic mutations can be detected by increasing the read depth using NGS, but this is not always true. When preparing the library, the amount of genomic DNA is the first thing to consider. A human cell contains about 6 pg of DNA. Therefore, 60 ng of DNA corresponds to 10^4^ cells. If 0.1% of the cells have a mosaic mutation, there will be just 10 copies of the mutated allele in 60 ng of DNA, and 0.01% of mosaics will have a single copy. Therefore, lower frequency mosaics are theoretically undetectable. The next problem is distinguishing between artifacts due to PCR errors during library preparation and real low-frequency mosaic variants. Previously, we examined dual sequencing of NGS and extracted common low-frequency mosaic calls, then removed common artifacts caused by sequence-dependent PCR errors [19]. This method had a detection sensitivity of 85.2% and a positive predictive value of 96.6% for the experimentally synthesized 10% mosaic, and 69.0% and 95.8% for the 1% mosaic. When the mosaic rate was 10% (VAF 5%) or more, the mutation peak could usually be confirmed by Sanger sequencing (Figure 5a). CEL nuclease-mediated heteroduplex incision with polyacrylamide gel electrophoresis and silver staining (CHIPS) was even more sensitive and could detect less than 1% of mosaic variants (Figure 5b). If target mutations have been determined by testing a proband or tumor, and if only the wild-type allele can be eliminated by digesting with specific restriction enzymes, mosaic mutations with lower frequencies can be detected. Starting with sufficient amounts of genomic DNA, repeated restriction enzyme digestion, and nested PCR enrichment for mutant alleles allow direct detection [28,29]. Empirically, this method is the most reliable for detecting very low-frequency mosaicism, but it cannot be used if the appropriate restriction enzymes are absent. Even if NGS predicts low-frequency mosaic mutations, there is currently no universal and simple confirmation method. In this study, only mosaic mutations that were finally detected by the Sanger method were counted, so the possibility that patients with low-frequency mosaic mutations remained among patients whose mutations could not be detected cannot be denied.

Regarding the phenotype of mosaic cases, among 13 patients with low-frequency mosaic mutations in blood DNA, only one had DD/ID (Fisher’s *p* = 0.0494 to definite TSC in this study), another one had epilepsy (Fisher’s *p* = 6.966×10^−5^), and four cases were familial (Appendix A). Neuropsychiatric symptoms tended to be milder, and the condition was accompanied by the chief complaint of organ lesions. This result was reasonable because, in the case of mosaic mutations, only cells with mutations produce symptoms.

### 3.6. Parental Mutation

Single-point mutation tests identified parental mutations in 13 of 49 families. Of the 13 families, only three had been diagnosed with TSC before genetic testing. This fact reveals that a significant proportion (approximately 20%) of the parents of patients not previously thought to have TSC then showed it. Therefore, genetic testing can efficiently diagnose phenotypically mild patients in families.

### 3.7. Analytical Method: Advantage of CoLAS

Mutation screening in this study was performed using the CHIPS-based method [18] in 188 patients, and using the newly developed CoLAS method [19] in the remaining 95 patients. Various advantages of the CoLAS method were observed.

#### 3.7.1. Mutation Detection Rate

Table 11 shows the overall mutation detection rate utilizing CHIPS and CoLAS after removing patients with large chromosomal-level deletions. In definite TSC patients, the mutation detection rate of CoLAS reached 87.5% and was significantly higher than the CHIPS-based screening method. In our previous study, all 138 variants in the *TSC1*/*2* gene detected by CHIPS and confirmed by Sanger sequencing were also detectable by CoLAS [19]. CoLAS also enabled the detection of mutations that CHIPS was unable to detect. In fact, of the 200 mutations detected in this study, 27 were mutations that could only be confirmed by CoLAS, including mosaic mutations, intragenic deletions, and splicing abnormalities due to deep intronic mutations (Appendix A). ColAS has the advantage of direct determination of breakpoint sequences for the intragenic deletions, and the presence of heterozygous SNPs can demonstrate mosaic deletions (Figure 6a).

#### 3.7.2. Consequence of Splicing Mutations

Until now, it has been challenging to determine the consequences of splicing mutations, but the development of the CoLAS method makes it possible to measure abnormal splicing events directly and semi-quantitatively (Figure 6b). The effects of intronic mutations affecting mRNA splicing are diverse, including activation of cryptic splice acceptor or donor sites, creating a new acceptor, donor or branch point site, and exon skip. A single DNA mutation can cause these splicing abnormalities singly or in various combinations, making it difficult to predict without demonstrating experimentally (Table 12, Figure 7).

### 3.8. Limitation of This Study

Patients in this study were recruited for genetic testing and were not a natural cohort. However, 225 out of 283 patients (79.5%) were clinically diagnosed with definite TSC, and the overall age distribution was the same as the TOSCA baseline data. Although this study included more infants than TOSCA, it is considered to reflect the actual distribution of TSC patients in Japan to some extent.

Epilepsy was less frequent in this study than in TOSCA, even after adjusting for age and causative genes. The prevalence of epilepsy in our study is consistent with previous studies targeting older Japanese patients [21], and it is likely that the diagnosis rate of TSC patients without epilepsy is actually higher in Japan. This study did not investigate in detail the types and severity of epilepsy. Although this study suggests that TSC1 patients may have a higher remission rate than TSC2 patients, it remains a possibility that a more detailed analysis could prove TSC2 patients have more severe epilepsy. For example, in the TOSCA study, the rate of infantile spasms was higher in TSC2 patients than in TSC1 patients (47.3% vs. 23%) [34].

This study did not address TAND symptoms. Patients were recruited from hospitals throughout Japan for the purpose of genetic testing, so it was difficult to implement standardized scales and tests for developmental disorders and psychiatric symptoms. DD/ID was used instead, as an index of psychiatric symptoms. Therefore, it is unknown what kind of TAND symptoms patients without intellectual disabilities have, and whether there are any differences between TSC1 and TSC2 patients.

## 4. Materials and Methods

### 4.1. Patient Collection

The patients in this study were clinically diagnosed or suspected of TSC. Samples were collected from all regions in Japan for clinical genetic testing of TSC, from 2002 to 2022. Among them, 53 patients were previously reported [18] and added to this study after re-evaluation. Written informed consent was obtained from all patients by their primary physician for the mutational and clinical data analysis. The study design was approved by the institutional review board of Kanazawa Medical University (G111, G160).

### 4.2. Clinical Evaluations

The clinical diagnosis of TSC was classified into definite TSC and possible TSC, according to the latest international diagnostic criteria [1]. Previous patients were also reassessed using these updated criteria. Clinical data of the patients were accumulated using the check sheet filled out by the primary physicians. This clinical check sheet was prepared with reference to Dabora et al. [12]. The check sheet assessed TSC diagnosis and scored the severity of the patient’s phenotype for several aspects. The scoring system is summarized in Table 13. A score of 0 to 3 was given, according to the presence, number, size, and severity of lesions and symptoms. We could not use the intellectual quotient (IQ) scale in infants and young patients to assess intellectual performance. Thus, we defined the criteria for “developmental delay and/or intellectual disability (DD/ID)” as follows: Score 0: Those who are normal and do not require an IQ test, or the IQ or developmental quotient (DQ), evaluated on any established scale, was 70 or more, and no obvious psychomotor developmental delay was observed in infants by a pediatric neurologist. Score1, 2, 3: IQ or DQ was between 50 and 69, 35 and 49, less than 34, respectively. Patients with intellectual disabilities with unknown severity were classified as score 2. Items of the scoring systems are included in the latest diagnostic criteria; however, this scoring system is not the criteria themselves. For the number of lesions it is necessary to take a positive finding in diagnostic criteria for some lesions. The columns shown in gray in Table 13 did not meet the diagnostic criteria. The color of each item in the score table is linked to Figure 3 and Appendix A.

### 4.3. Mutation Analysis

#### 4.3.1. Mutation Screening Test

*TSC1* and *TSC2* gene analysis was mainly performed using patient peripheral blood. A total of 188 patients were analyzed using CEL nuclease-mediated heteroduplex incision with polyacrylamide gel electrophoresis and silver staining (CHIPS) [48,49,50] and long-range PCR, as previously reported [18]. CHIPS is an optimized gel shift assay-based method for the enzyme mismatch cleavage. Heteroduplex analysis and mismatch cleavage analysis can be performed simultaneously, and since it is more sensitive than Sanger sequencing, it is also useful for detecting mosaic mutations. The remaining 95 patients were analyzed using our newly developed method, called combined long amplicon sequencing (CoLAS) [19,20]. CoLAS is a targeted DNA and RNA sequencing, using long-range PCR-based NGS to detect diverse mutations in *TSC1* and *TSC2*. Simultaneous analysis of the entire genomic region of the genes including introns, and full-length cDNA splicing analysis using the SMARTer method (SMART-Seq^®^ HT Kit, Takara Bio, Mountain View, CA, USA), CoLAS can detect splicing abnormalities due to deep intronic mutations and complex intragenic large deletions, including mosaic mutations.

#### 4.3.2. Sanger Sequencing

To validate the CHIPS and CoLAS results, and the *TSC1* and *TSC2* for each exon, PCR was performed as previously reported [18], and direct DNA sequencing was performed using a BigDye Terminator v3.1 cycle sequencing kit and ABI PRISM 3100 xl Genetic analyzer (Thermo Fisher Scientific, Waltham, MA, USA).

#### 4.3.3. Confirm Chromosomal Level Large Deletion

CoLAS can detect heterozygous single nucleotide polymorphisms (SNPs), including intron regions. With an absence of heterozygous SNPs over the entire region of *TSC2* (approximately 40 Kb) or *TSC1* (approximately 56 Kb), a large chromosomal-level deletion of that region was assumed. CytoScan 750K array or CytoScan HD array (Thermo Fisher Scientific) were used to confirm the chromosomal level large deletion. *TSC2*/*PKD1* contiguous gene deletion was correctly diagnosed in this way.

#### 4.3.4. Mosaic Mutation Analysis

A diagnosis of mosaic mutation in a proband was considered positive only when confirmed by Sanger sequencing or CHIPS. A method of eliminating the wild-type allele with restriction enzymes and nested PCR [28] and a method of excising the heteroduplex bands from CHIPS gel and re-amplification were effective for confirming low-frequency mosaic mutations. For the diagnosis of low-frequency mosaicism in the parent of the proband, when these methods could not be used, deep sequencing was performed twice on the approximately 1-kb PCR product containing the mutation point. Then it was judged using GATK’s Mutect2 (Version 4.0.6.0, Broad institute, Cambridge, MA, USA) [51], according to the method described previously [19].

#### 4.3.5. Interpretation of Variants

Whether or not the detected variants had pathological significance was determined according to the guideline of the American college of medical genetics and genomics (ACMG) 2015 [52]. Only “pathogenic” or “likely pathogenic” variants were identified as disease-causable mutations.

### 4.4. Statistical Analysis

Differences in the frequency of symptoms and lesions between the TOSCA baseline data and this study were analyzed using Pearson’s chi-square test. Differences between TSC1 and TSC2 patients in this study were analyzed using Fisher’s exact test. In both tests, *p* < 0.05 (double-tailed) was considered statistically significant.

### 4.5. DATA Deposition

All *TSC1*/*TSC2* mutations described in this paper were registered in ClinVar (https://www.ncbi.nlm.nih.gov/clinvar/, last accessed on 15 August 2022), and the submission IDs are indicated in Appendix A.

## 5. Conclusions

Lesions of tuberous sclerosis are divided into three types: those that appear in the fetal period and remain unchanged after birth, those that appear from childhood to adulthood, and those that appear later, mainly in adulthood. The first is hamartia, represented by cortical tubers and HM, which is thought to be due to haploinsufficiency; the next is AF and renal AML, and these hamartomas have LOH; and the last is represented by LAM, which has LOH, and some metastatic lesions from renal AMLs are included. Retinal hamartomas were generally TSC2-specific lesions. MMPH seemed to be TSC1 dominant lesion, but this requires further investigations. The prevalence of UF and cardiac rhabdomyoma was higher in TSC1 and TSC2 patients, respectively. TSC2 patients showed a more severe phenotype from high DD/ID frequency, tough AF and renal AML lesions, and non-remission of epilepsy. *TSC2* PP mutations may attenuate AF and renal AML compared with NP mutations. CoLAS had an advantage in detecting broad *TSC* mutations. Approximately 20% of parents of the proband showed mild TSC that might have been missed. The patient demographics presented in this study had a high frequency of patients with TSC1 and a low prevalence of epilepsy. The reason why many patients with mild neuropsychiatric symptoms were diagnosed seems to be the high prevalence of brain imaging in Japan, suggesting the possibility that mild TSC patients might not be correctly diagnosed worldwide.

## Figures and Tables

**Figure 1 ijms-23-11175-f001:**
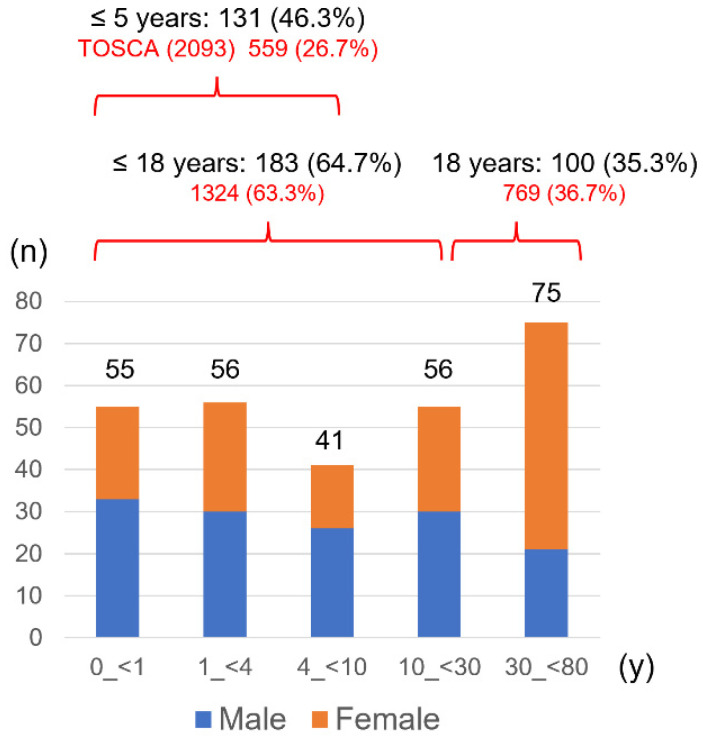
Patient distribution by age in this study (*n* = 283).

**Figure 2 ijms-23-11175-f002:**
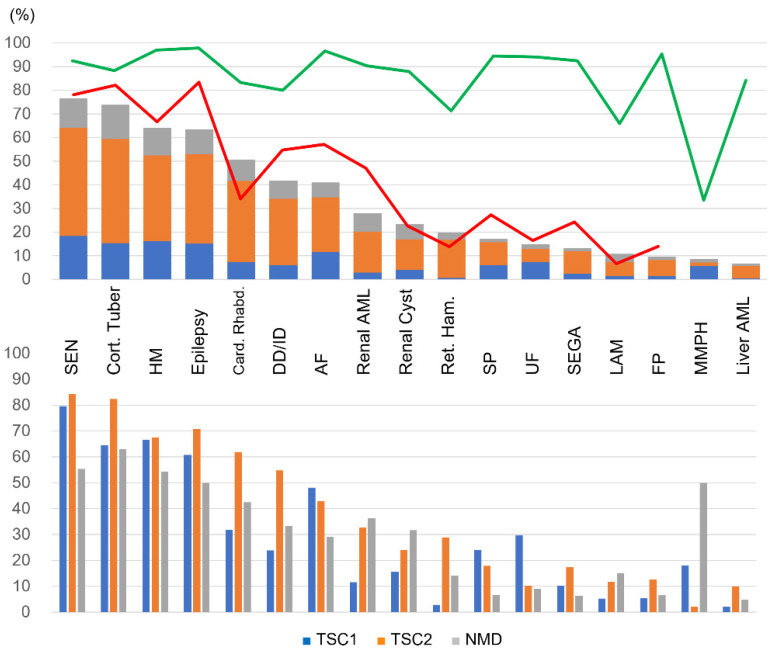
Frequency of each lesion and symptom. (**upper panel**) Frequency of each lesion among all definite TSC patients. The proportion of TSC1, TSC2, and no mutation detected (NMD) for each symptom is shown. The green line indicates the testing rates for each condition. The red line indicates the frequency in the TOSCA baseline study. (**lower panel**) Percentage of each symptom by TSC1, TSC2, and NMD patients are presented. AF: facial angiofibroma; AML: angiomyolipoma; Card. Rhabd.: cardiac rhabdomyoma; Cort.Tuber: cortical tubers; DD/ID: developmental delay/intellectual disability; FP: forehead plaque; HM: hypomelanotic macules; LAM: lymphangiomyomatosis; MMPH: multifocal micronodular pneumocyte hyperplasia; Ret. Ham.: retinal hamartoma; SEGA: subependymal giant cell astrocytoma; SEN: subependymal nodules; SP: shagreen patch; UF: ungual fibroma.

**Figure 3 ijms-23-11175-f003:**
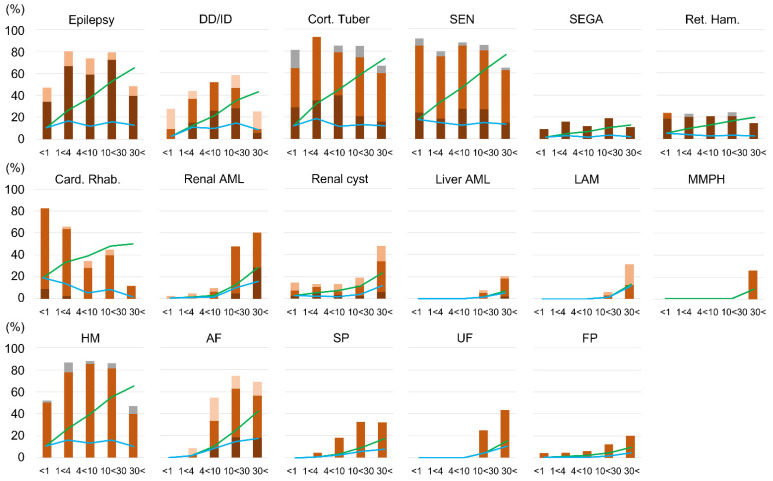
Frequency of each lesion and symptom in each age range of definite TSC (*n* = 225). The color of each bar indicates the score in Table 13. The green line indicates the cumulative percentage, and the blue line indicates the percentage increase from previous age groups. Abbreviations are the same as in Figure 2.

**Figure 4 ijms-23-11175-f004:**
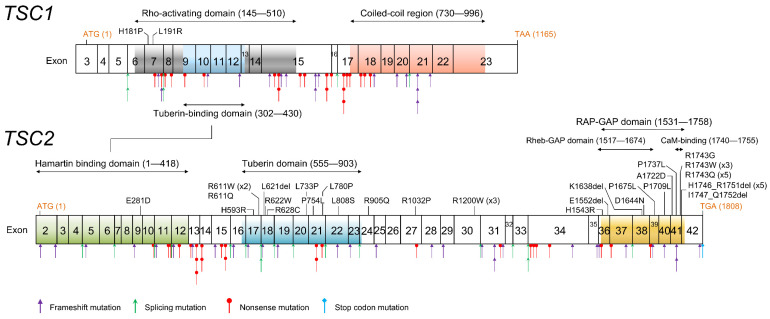
Distribution of *TSC1* and *TSC2* point mutations. Missense and in-frame mutations are indicated in the upper lane of each gene’s schema. In contrast to the protein truncated mutations diffusely distributed in both *TSC1* and *TSC2*, missense and in-frame mutations are concentrated in functional domains.

**Figure 5 ijms-23-11175-f005:**
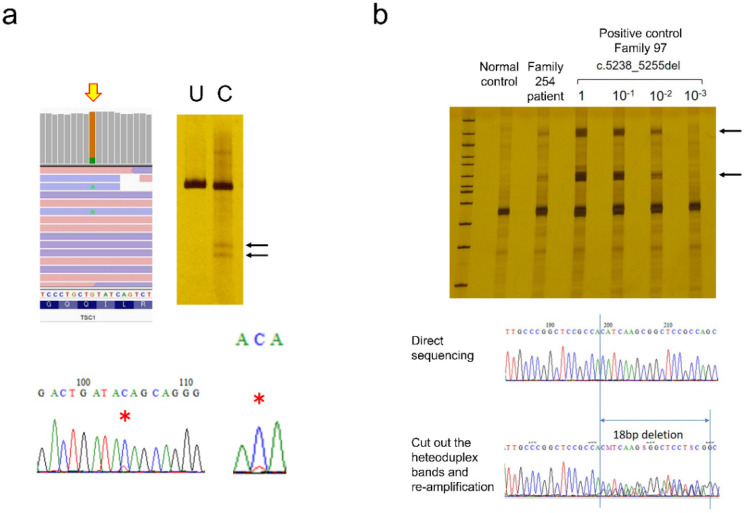
Examples of mosaic analysis. (**a**) *TSC1* mosaic mutation, NM_000368.5:c.1960C>T p.(Gln654Ter), detected in family 267. CHIPS and Sanger sequencing detected this mosaic mutation (black arrows and red asterisks). (**b**) *TSC2* very low-frequency mosaic mutation detected in family 254. NGS analysis (CoLAS) predicted *TSC2* 18bp deletion as a mosaic. Direct sequencing could not detect the mutation. As family 97 has the same mutation, a dilution series was made with control DNA, and CHIPS analysis was performed. When the heteroduplex bands (black arrows) were excised and sequenced, the 18 base pair deletion could be detected in the patient.

**Figure 6 ijms-23-11175-f006:**
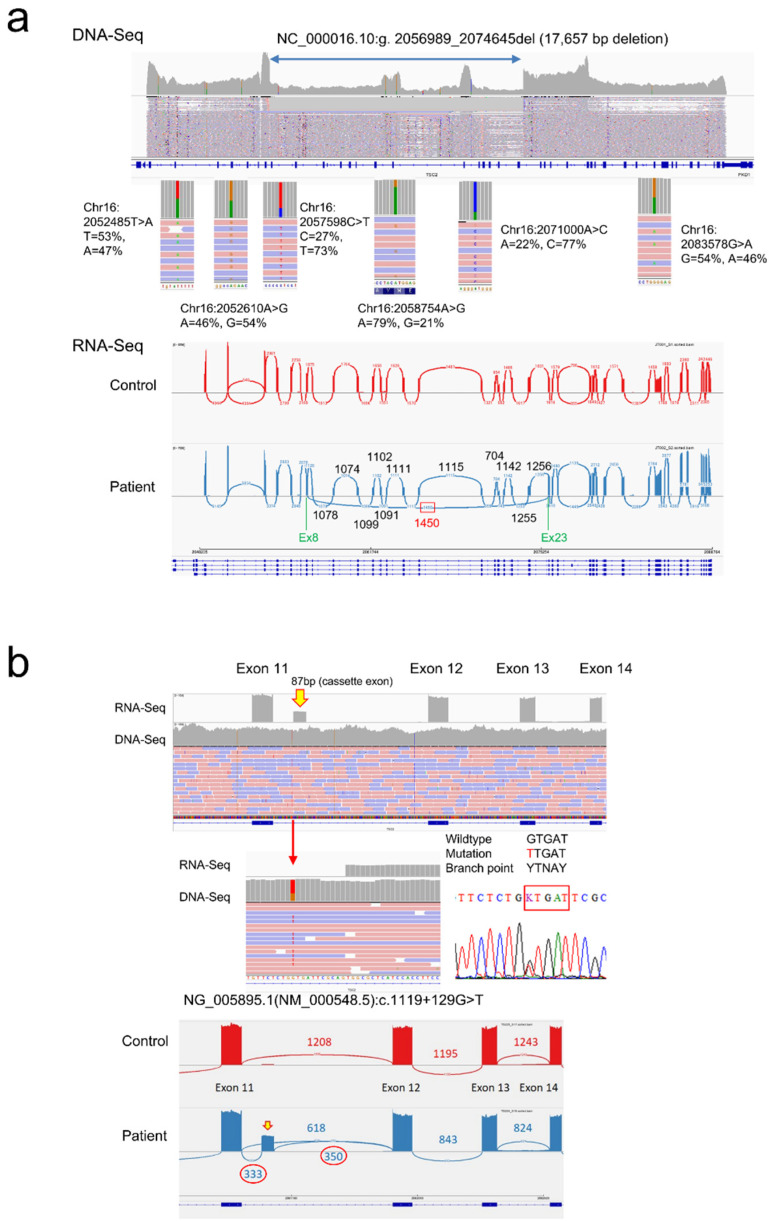
Examples of CoLAS analysis. (**a**) A *TSC2* intragenic large deletion was detected in family J2. A large deletion of 17657 bp was confirmed at the DNA level. The deletion site contains heterozygous SNPs, the allelic ratio was biased inside the deletion, and almost 1:1 outside the deletion, indicating that this large deletion was a mosaic mutation. Corresponding to the large deletion, an abnormal junction from exon 8 to 23 was observed at the RNA level. (**b**) A *TSC2* deep intronic mutation was detected in family 224. A single base substitution in intron 11 created a new branch point and inserted an 87 base cassette exon. A stop codon appeared within the cassette exon, resulting in a protein-truncating mutation.

**Figure 7 ijms-23-11175-f007:**
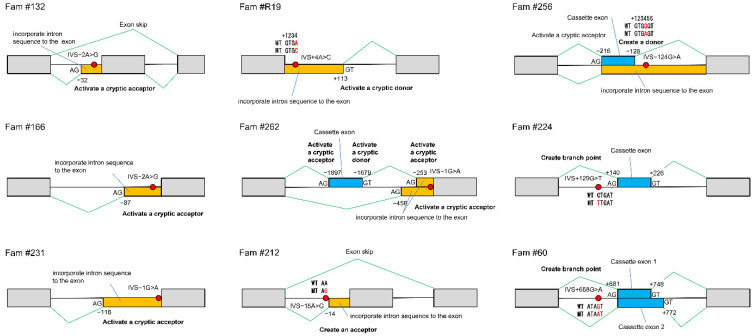
The consequences of complex splicing mutations. WT: wildtype allele sequence; MT: mutation allele sequence.

**Table 1 ijms-23-11175-t001:** Demographics of the participants (*n* = 283).

Characteristics	Data		
Age (year)	All	*TSC1* mutation positive	*TSC2* mutation positive
average/median/range	16.4/7/0–80	21.4/21/0–77	11.4/4/0–80
Heredity	Familial	Sporadic	Proband
*n* (%), family	44 (15.5), 21	239 (84.5)	260 (91.8)
Gender	Male	Female	Unknown
n (%)	140 (49.5)	142 (50.2)	1 (0.3)
Clinical diagnosis	Definite TSC	Possible TSC	Not-examined-well (Genetic diagnosis)
*n* (%)	225 (79.5)	53 (18.7)	5 (1.8)

**Table 2 ijms-23-11175-t002:** Comparison of symptom and lesion prevalence between the TOSCA baseline, a previous Japanese study, and this study.

	TOSCA	Ref [21]	Definite TSC in This Study							
(*n*)	Base Line (2093)	Wataya-Kaneda 2013 (166) ^#^	All Age (225)		Age ≥ 10 (99)		*TSC2* (118)		*TSC1* (56)	
Lesions	*n* (%)	%	*n* (%)	*p **	*n* (%)	*p **	*n* (%)	*p **	*n* (%)	*p **
Epilepsy	1748 (83.5)	63	225 (64.0)	0.0000	99 (61.6)	0.0000	118 (71.2)	0.0009	56 (58.9)	0.0000
DD/ID	822 (54.9)	42	182 (41.8)	0.0018	99 (39.4)	0.0050	91 (54.9)	1.0000	46 (26.1)	0.0003
Cort. Tuber	1721 (82.2)	ND	202 (73.8)	0.0048	84 (66.7)	0.0006	106 (83.0)	0.9378	48 (64.6)	0.0034
SEN	1636 (78.2)	77	212 (76.9)	0.7344	89 (70.8)	0.1328	113 (85.0)	0.1139	49 (79.6)	0.9517
SEGA	510 (24.4)	2	212 (13.2)	0.0012	89 (14.6)	0.0422	113 (16.8)	0.1115	49 (12.2)	0.0831
Ret. Ham.	294 (14.0)	ND	161 (19.9)	0.1299	64 (17.2)	0.6374	88 (27.3)	0.0058	35 (5.7)	0.2711
HM	1399 (66.8)	65	223 (64.6)	0.5568	98 (58.2)	0.1001	118 (66.9)	1.0000	54 (66.7)	1.0000
AF	1199 (57.3)	93	222 (41.4)	0.0000	98 (71.4)	0.0087	117 (43.6)	0.0059	54 (48.1)	0.2350
SP	573 (27.4)	83	217 (17.5)	0.0054	93 (32.3)	0.3994	115 (18.3)	0.0542	54 (24.1)	0.7148
UF	350 (16.7)	64	216 (15.3)	0.7724	93 (35.3)	0.0001	115 (10.4)	0.1481	54 (29.6)	0.0340
FP	295 (14.1)	46	219 (10.0)	0.1977	96 (16.7)	0.6771	116 (12.9)	0.6924	55 (5.5)	0.1171
Card. Rhabd.	717 (34.3)	46	190 (50.5)	0.0001	72 (29.2)	0.4541	104 (61.5)	0.0000	44 (31.8)	0.8615
Renal AML	987 (47.2)	61	207 (28.0)	0.0000	95 (54.7)	0.1955	108 (32.4)	0.0046	52 (13.5)	0.0000
Renal Cyst	477 (22.8)	28	201 (23.4)	0.9598	92 (34.8)	0.0217	106 (24.5)	0.8080	51 (15.7)	0.3204
LAM	144 6.9)	39	148 (11.5)	0.2553	80 (21.3)	0.0033	75 (12.0)	0.3133	38 (5.3)	0.9968
Age range (Yr.)	0–71	0–78	0–80							
Average	ND	26.6	16.4							
Median	13	ND	7							

ND: not described in the literature, ^#^ the number of patients evaluated for each lesion is not shown in the literature, * Pearson’s chi-squared test for TOSCA, abbreviations are the same as in Figure 2.

**Table 3 ijms-23-11175-t003:** Mutation detection rate.

	Definite TSC	Possible TSC	NEW	Total
	*n* (%)	*n* (%)	*n* (%)	*n* (%)
*TSC1*	56 (24.9)	6 (11.3)	1 (20.0)	63 (22.3)
*TSC1*/*TSC2 **	1 (0.4)			1 (0.4)
*TSC2*	113 (50.2)	15 (28.3)	3 (60.0)	131 (46.3)
*TSC2*/*PKD1*	4 (1.8)			4 (1.4)
NMD	51 (22.7)	32 (60.4)	1 (20.0)	84 (29.7)
Mutation detected	174 (77.3)	21 (39.6)	4 (80.0)	199 (70.3)
total	225	53	5	283

* One patient had both *TSC1* and *TSC2* mutations. NEW: not-evaluated-well; NMD: no mutation detected; *TSC2*/*PKD1*: *TSC2*/*PKD1* contiguous deletion syndrome.

**Table 4 ijms-23-11175-t004:** Distribution of mutation types.

	*TSC1*	*TSC2*	Total
	*n* (%) All/Proband	*n* (%) All/Proband	*n* (%) All/Proband
Mutations predicted to produce protein			
Missense	3/2 (4.7/3.8)	* 35/* 31 (25.7/24.6)	38/33 (19.0/18.4)
In-frame	0	11/10 (8.1/7.9)	11/10 (5.5/5.6)
Stop codon	0	1/1 (0.7/0.8)	1/1 (0.5/0.6)
subtotal	3/2 (4.7/3.8)	47/42 (34.6/33.3)	50/44 (25.0/24.6)
Mutations predicted to produce truncated protein or no protein			
Frameshift	18/16 (28.1/30.2)	33/30 (24.3/23.8)	51/46 (25.5/25.7)
Nonsense	30/26 (46.9/49.1)	27/27 (19.9/21.4)	57/53 (28.5/29.6)
Splicing	* 11/* 7 (17.2/13.2)	20/19 (14.7/15.1)	31/26 (15.5/14.5)
Intragenic large deletion	2/2 (3.1/3.8)	5/5 (3.7/4.0)	7/7 (3.5/3.9)
*TSC2*/*PKD1* deletion		4/3 (2.9/2.4)	4/3 (2.0/1.7)
subtotal	61/51 (95.3/96.2)	89/84 (65.4/66.6)	150/135 (75.0/75.4)
total	* 64/* 53	* 136/* 126	* 200/* 179

* Counting both a *TSC1* splicing mutation and a *TSC2* missense mutation from one patient.

**Table 5 ijms-23-11175-t005:** *TSC1*/*TSC2* mutation ratio comparing between this study and previous large scale mutation studies.

Studies	*TSC1* Mutation				*TSC2* Mutation				*TSC1*/*TSC2* Ratio in All Cases		*TSC1*/*TSC2* Ratio in Sporadic Cases	
	F	S	U	Total	F	S	U	Total		*p **		*p **
This study ^#^	16	40		56	16	101		117	0.48		0.40	
TOSCA [17]			178	178			571	571	0.31	0.0255		
US [11]	11	5		16	16	42		58	0.28	0.0950	0.12	0.0169
US [12]	6	22		28	29	129		158	0.18	0.0002	0.17	0.0043
US [13]	25	36		61	28	154		182	0.34	0.1214	0.23	0.0482
UK [24]	9	13		22	9	88	1	98	0.22	0.0287	0.15	0.0044
Denmark [25]	2	2	7	11	6	17	17	40	0.28	0.5191	0.12	0.1620
Netherland [26]			82	82			280	280	0.29	0.0200		
Taiwan [27]	2	7		9	8	47		55	0.16	0.0051	0.15	0.0254

F: familial; S: sporadic; U: unknown. * Fisher’s exact test. ^#^ Includes only definite TSC for comparison to other studies. *TSC2/PKD1* contiguous deletion syndrome are included in *TSC2* mutation, and a patient with both *TSC1* and *TSC2* mutations is excluded.

**Table 6 ijms-23-11175-t006:** Difference in each symptom and lesion between TSC1 and TSC2 patients.

Subject	All Patients			Definite TSC			Age ≥ 10 Yr.			Age ≥ 20 Yr.		
Mutated Gene	*TSC1*	*TSC2*		*TSC1*	*TSC2*		*TSC1*	*TSC2*		*TSC1*	*TSC2*	
*n*	62	128		56	113		36	42		31	26	
Av. age (yr.)	20.5	10.5		20.4	10.7		33.4	27.9		36.6	36.2	
Med. age (yr.)	19.5	3		19.5	3		31.5	26.5		33	32.5	
Lesion	*n* (%) *	*n* (%)	*p* ^#^	*n* (%)	*n* (%)	*p* ^#^	*n* (%)	*n* (%)	*p* ^#^	*n* (%)	*n* (%)	*p* ^#^
Epilepsy	62 (56.5)	128 (66.4)	0.2016	56 (58.9)	113 (70.8)	0.1645	36 (55.6)	42 (73.8)	0.1019	31 (54.8)	26 (61.5)	0.7882
DD/ID	51 (23.5)	97 (57.7)	0.0000	46 (23.9)	87 (58.6)	0.0002	36 (25.0)	42 (66.7)	0.0003	31 (32.3)	26 (42.3)	0.6255
Cort. Tuber	42 (73.8)	111 (79.3)	0.5155	41 (75.6)	98 (86.7)	0.1339	22 (77.3)	36 (72.2)	0.7639	21 (61.9)	21 (61.9)	1.0000
SEN	47 (83.0)	118 (81.3)	1.0000	45 (86.7)	106 (89.6)	0.5843	26 (84.6)	37 (81.1)	1.0000	22 (81.8)	22 (72.7)	0.7205
SEGA	51 (9.8)	122 (18.9)	0.1769	49 (10.2)	110 (18.2)	0.2442	26 (15.4)	39 (25.6)	0.3733	21 (14.3)	23 (17.4)	1.0000
Ret. Ham.	35 (5.7)	100 (25.0)	0.0138	35 (5.7)	87 (28.7)	0.0069	17 (11.8)	30 (26.7)	0.2894	14 (14.3)	16 (25.0)	0.6567
HM	60 (63.3)	127 (62.2)	1.0000	54 (66.7)	113 (68.1)	0.8611	35 (57.1)	42 (64.3)	0.6398	30 (50.0)	26 (53.8)	0.7952
AF	59 (44.1)	127 (37.8)	0.4528	54 (48.1)	112 (42.0)	0.5059	34 (70.6)	42 (76.2)	0.6098	29 (72.4)	26 (73.1)	1.0000
SP	59 (22.0)	128 (16.0)	0.4102	54 (48.1)	111 (18.0)	0.4087	33 (33.3)	40 (37.5)	0.8079	28 (35.7)	25 (28.0)	0.5722
UF	60 (30.0)	125 (8.8)	0.0004	54 (29.6)	111 (9.9)	0.0029	31 (58.1)	40 (27.5)	0.0146	29 (58.6)	26 (30.8)	0.0579
FP	60 (5.0)	127 (11.0)	0.2757	55 (5.5)	112 (12.5)	0.1845	34 (8.8)	41 (22.0)	0.2051	29 (10.3)	26 (26.9)	0.1644
Card. Rhabd.	46 (34.8)	114 (61.4)	0.0028	44 (31.8)	102 (62.7)	0.0010	22 (18.2)	34 (41.2)	0.0868	17 (5.9)	18 (22.2)	0.3377
Renal AML	54 (11.1)	116 (29.3)	0.0110	52 (11.5)	104 (32.7)	0.0220	30 (20.0)	40 (75.0)	0.0000	25 (24.0)	25 (76.0)	0.0005
Renal cyst	53 (15.1)	113 (21.2)	0.4043	51 (15.7)	101 (22.8)	0.3952	29 (24.1)	39 (38.5)	0.2957	24 (25.0)	23 (47.8)	0.1351
Liver AML	52 (1.9)	107 (9.3)	0.1039	50 (2.0)	95 (10.5)	0.0976	28 (3.6)	36 (27.8)	0.0169	23 (8.7)	21 (38.1)	0.0310
LAM	38 (5.3)	83 (10.8)	0.4993	38 (5.3)	72 (11.1)	0.4891	25 (8.0)	33 (27.3)	0.0930	21 (9.5)	23 (34.8)	0.0725
MMPH	23 (21.7)	46 (2.2)	0.0120	22 (18.2)	43 (2.3)	0.0413	17 (29.4)	19 (5.3)	0.0807	16 (31.3)	14 (7.1)	0.1755

Comparing the patients with *TSC1* and *TSC2* mutations. Patient with both *TSC1* and *TSC2* mutations, patients with *TSC2/PKD1* contiguous gene deletion syndrome, and clinically not well evaluated patients were excluded. * Each number indicates an evaluated patient number (positive rate in %), ^#^ Fisher’s exact test, Av.: average; Med.: median.

**Table 7 ijms-23-11175-t007:** Proportion of severe lesions between TSC1 and TSC2 patients.

Subject *		All Patients			Age ≥ 10 Yr.		
Mutated Gene		*TSC1*	*TSC2*	*p* ^#^	*TSC1*	*TSC2*	*p* ^#^
Lesion	Score ^$^	*n* (%) ^$^	*n* (%)		*n* (%)	*n* (%)	
Epilepsy	3	62 (46.8)	128 (53.9)	0.4391	36 (44.4)	42 (69.0)	0.0389
DD/ID	3	51 (3.9)	97 (28.9)	0.0002	36 (2.8)	42 (33.3)	0.0005
SEN	3	51 (17.6)	123 (26.0)	0.3264	27 (25.9)	38 (26.3)	1.0000
Cort. Tuber	3	50 (16.0)	116 (35.3)	0.0154	26 (15.4)	36 (25.0)	0.5291
AF	3	59 (3.4)	128 (12.5)	0.0616	34 (5.9)	42 (33.3)	0.0042
Renal AML	2 or 3	54 (11.1)	116 (26.7)	0.0273	30 (20.0)	40 (75.0)	0.0000
Renal AML	3	54 (1.9)	116 (9.5)	0.1060	30 (3.3)	40 (27.5)	0.0095
Renal cyst	2 or 3	53 (11.3)	113 (12.4)	1.0000	29 (20.7)	39 (23.1)	1.0000
Liver AML	2 or 3	52 (1.9)	107 (7.5)	0.2732	28 (3.6)	36 (22.2)	0.0656

* Subjects are the same as in Table 6. ^$^ See Table 13, ^#^ Fisher’s exact test, ^$^ each number indicates an evaluated patient number (percentage of patients with the specified score).

**Table 8 ijms-23-11175-t008:** Difference of each symptom and lesion between *TSC2* mutations predicted to produce a protein (PP), and mutations predicted to produce truncated protein or no protein (NP).

Subject	All Patients			Age ≥ 4 Yr.		
Class	PP	NP		PP	NP	
*n*	41	73		19	33	
Av. age (yr.)	7.8	10.3		15.9	21.7	
Med. age (yr.)	3	3		8	19	
	*n* (%) *	*n* (%)	*p* ^#^	*n* (%)	*n* (%)	*p* ^#^
Epilepsy	41 (65.8)	73 (76.7)	0.2731	19 (89.5)	33 (87.9)	1.0000
DD/ID	28 (64.2)	56 (64.3)	1.0000	17 (82.4)	33 (75.8)	1.0000
SEN	35 (71.4)	71 (94.4)	0.0019	18 (72.2)	32 (96.9)	0.0432
SEGA	39 (15.4)	72 (20.8)	0.6143	18 (22.2)	32 (21.9)	1.0000
Cort. tuber	34 (76.5)	65 (89.2)	0.1382	16 (75.0)	31 (87.1)	0.6667
Ret. Ham.	28 (21.4)	59 (30.5)	0.4484	13 (23.1)	26 (30.8)	0.7144
HM	41 (65.8)	73 (65.8)	1.0000	19 (78.9)	33 (75.8)	1.0000
AF All	41 (26.8)	72 (45.8)	0.0703	20 (55.0)	33 (93.9)	0.0012
AF S2+3	41 (17.1)	72 (36.1)	0.0343	20 (40.0)	33 (75.8)	0.0181
AF S3	41 (4.9)	72 (19.4)	0.0474	20 (10.0)	33 (42.4)	0.0150
SP	24 (12.5)	71 (22.5)	0.3834	17 (17.6)	32 (46.9)	0.0631
UF	38 (0)	72 (13.9)	0.0144	16 (0.0)	33 (30.3)	0.0197
FP	40 (5.0)	72 (15.3)	0.1312	18 (11.1)	33 (21.2)	0.4642
Card. Rhabd.	39 (64.1)	65 (64.6)	1.0000	19 (42.1)	28 (42.9)	1.0000
Renal AML All	38 (18.4)	65 (36.9)	0.0739	18 (38.9)	32 (66.8)	0.0720
Renal AML S2 + 3	38 (15.8)	65 (33.8)	0.0658	18 (33.3)	32 (68.8)	0.0202
Renal AML S3	38 (2.6)	65 (12.3)	0.1490	18 (5.6)	32 (25.0)	0.1304
Renal cyst	38 (15.7)	63 (25.4)	0.3239	18 (22.2)	30 (36.7)	0.3512
Liver AML	36 (5.6)	58 (12.1)	0.4746	17 (11.8)	29 (24.1)	0.4503
LAM	24 (8.3)	50 (10.0)	1.0000	10 (20.0)	26 (19.2)	1.0000

We excluded a patient with both *TSC1* and *TSC2* mutations, those with *TSC2*/*PKD1* contiguous gene deletion syndrome, and those with mosaic mutation. * Each number indicates an evaluated patient number (percentage of patients with the specified score). ^#^ Fisher’s exact test, Av.: average; Med.: median; NP: *TSC2* mutations predicted to produce a truncated protein or no protein, including frameshift, nonsense, splicing, and large deletions.; PP: TSC2 mutations are predicted to produce a protein, including missense, in-frame, and stop codon mutations.; S2 + 3: score 2 or 3; S3: score 3 in Table 13.

**Table 9 ijms-23-11175-t009:** Mosaic mutations detected in the patients with TSC.

Detection *	# Family	Gene	Variant (HGVS Format)	Mutation Type	Estimated VAF	Reference
1	10	*TSC1*	NG_012386.1(NM_000368.5):c.2209-258_2502+101del NG_012386.1(NM_000368.5):c.2041+240_2392-319del	Mosaic of two large deletions	NA	[19]
1	267	*TSC1*	NM_000368.5:c.1960C>T p.(Gln654Ter)	Nonsense	0.15	
1	254	*TSC2*	NM_000548.5:c.5238_5255del p.(His1746_Arg1751del)	In-frame	0.005	
1	R176	*TSC2*	NG_005895.1(NM_000548.5):c.2546-313_2967-376del	Large deletion	NA	
1	J2	*TSC2*	NC_000016.10:g.2056989_2074645=/2056989_2074645del	Large deletion	NA	
1	28	*TSC2*	NM_000548.5:c.2261C>T p.(Pro754Leu)	Missense	0.256	
1	229	*TSC2*	NM_000548.5:c.5126C>T p.(Pro1709Leu)	Missense	0.065	
1	R129	*TSC2*	NM_000548.5:c.1327C>T p.(Gln443Ter)	Nonsense	0.113	
1	271	*TSC2*	NM_000548.5:c.1492G>T p.(Glu498Ter)	Nonsense	0.34	
1	234	*TSC2*	NG_005895.1(NM_000548.5):c.[4570-18_4570-6del;4570-3_4570-2dup]	Splicing	0.13	
2	83	*TSC2*	NM_000548.5(TSC2):c.1981_2020del p.(Gly661LeufsTer24)	Frameshift	0.001	
2	204	*TSC2*	NM_000548.5:c.4798_4804dup p.(Glu1602delinsValTrpTer)	Frameshift	0.049	
2	270	*TSC2*	NM_000548.5:c.2764_2765del p.(Leu922ValfsTer3)	Frameshift	0.025	
3	SS	*TSC2*	NM_000548.5:c.5228G>A p.(Arg1743Gln)	Missense	ND in PB	[28]
3	191	*TSC2*	NM_000548.5:c.1372C>T p.(Arg458Ter)	Nonsense	ND in PB	[29]
3	231	*TSC2*	NM_000548.5:c.1840-1G>A r.1839_1840ins1840-116_1840-1 p.Ala614GlnfsTer15	Splicing	0.006	[29]
3	258	*TSC2*	NM_000548.5:c.1840-1G>A	Splicing	ND in PB	

* How it was detected: 1. detected in proband; 2. single-site testing in parents for mutation in proband; 3. mutations were detected in tumors, followed by those detected in the blood. NA: not applicable; ND in PB: not detected in peripheral blood.

**Table 10 ijms-23-11175-t010:** Parental mutation analysis.

Parental Mutation Was Detected (*n* = 13)								
# Family	Parents	Age (y)	Gene	Variant (HGVS Format)	Mutation Type	Mosaic	Clin. Dx before Proband Dx	Clin. Dx after Evaluation
32	Mother	35	*TSC1*	NM_000368.5:c.2626-1G>A	Splicing		Definite	Definite
36	Mother	68	*TSC1*	NM_000368.5:c.2582del p.(Leu861ProfsTer17)	Frameshift		Possible	Definite
69	Mother	57	*TSC1*	NG_012386.1(NM_000368.5):c.363+668G>T	Splicing		NEW	Possible
80	Father	28	*TSC1*	NM_000368.5:c.2347C>T p.(Gln783Ter)	Nonsense		NEW	Possible
186	Father	36	*TSC1*	NM_000368.5:c.1525C>T p.(Arg509Ter)	Nonsense		Possible	Definite
203	Mother	77	*TSC1*	NM_000368.5:c.2672dup p.(Asn891LysfsTer13)	Frameshift		NEW	NEW
212	Father	38	*TSC1*	NC_000009.12(NM_000368.5):c.664-15A>G	Splicing		NEW	Possible
83	Father	29	*TSC2*	NM_000548.5:c.1981_2020del p.(Gly661LeufsTer24)	Frameshift	Y	NEW	Possible
107	Father	33	*TSC2*	NM_000548.5:c.4930G>A p.(Asp1644Asn)	Missense		NEW	NEW
204	Mother	38	*TSC2*	NM_000548.5:c.4798_4804dup p.(Glu1602delinsValTrpTer)	Frameshift	Y	NEW	Definite
262	Father	29	*TSC2*	NG_005895.1(NM_000548.5):c.1717-1G>A	Splicing		NEW	NEW
270	Father	31	*TSC2*	NM_000548.5:c.2764_2765del p.(Leu922ValfsTer3)	Frameshift	Y	NEW	NEW
198	Mother	41	*TSC2*/*PKD1*	NC_000016.10:g.1851807_2093151del	Large deletion		NEW	Definite
Parental mutation was not detected (*n* = 36)								
# Family: 49,58, 88, 103, 108, 111, 112, 116, 117, 119, 122, 123, 127, 130, 135, 147, 159, 177, 185, 187, 201, 208, 213, 217, 221, 222, 224, 232, 234, 237, 243, 261, 263, 271, 272, 274.								

Clin. Dx: TSC clinical diagnosis; NEW: not examined well.

**Table 11 ijms-23-11175-t011:** Mutation detection rate by screening method.

	CHIPS	CoLAS	*p* ^#^
All patients	124/185 (67.0)	69/92 (75.0)	0.2118
Definite TSC	112/155 (72.3)	56/64 (87.5)	0.0145
Possible TSC	11/28 (39.3)	10/25 (40.0)	1.0000

Numbers in each item indicate mutation detected/examined (%), ^#^ Fisher’s exact test.

**Table 12 ijms-23-11175-t012:** Complex consequences of splicing mutations.

	DNA Level	RNA Level	Protein level (Predicted)	Mutation Type	Consequence	Reference
# TSC1 Family	NG_012386.1(NM_000368.5)	NM_000368.5	NP_000359.1			
69	c.363+668G>T	r.363_364ins363+681_363+748	p.Lys121_Met122ins[*38]	Branch point mutation	Creates two aberrant cassette exons in the intron.	[38]
		r.363_364ins363+681_363+772	p.Lys121_Met122ins[*30]			
132	c.664-2A>G	r.663_664ins[664-32_664-3;gg]	p.Met223PhefsTer12	An acceptor site mutation	Activates a cryptic acceptor site and incorporate intron sequence to 5′side of the exon.	
		r.664_737del	p.Pro222Val fsTer8		Exon skip	
166	c.364-2A>G	r.363_364ins[364-87_364-3;gg]	p.Met122CysfsTer18	An acceptor site mutation	activates a cryptic acceptor site and incorporates intron sequence to the 5′side of the exon.	
212	c.664-15A>G	r.665_737del	p.Pro222ValfsTer8	Create a new acceptor site	Exon skip	[19]
		r.664_665ins665-14_665-1	p.Met223AsnfsTer6		activates a cryptic acceptor site and incorporates an intron sequence to the 5′side of the exon.	
# TSC2 family	NG_005895.1(NM_000548.5)	NM_000548.5	NP_000539.2			
224	c.1119+129G>T	r.1119_1120ins1119+140_1119+226	p.Thr374TrpfsTer12	Branch point mutation	Creates an aberrant cassette exon in the intron.	Figure 6b
231	c.1840-1G>A	r.1839_1840ins[1840-116_1840-2;g>a]	p.Ala614GlnfsTer15	An acceptor site mutation	Activates a cryptic acceptor site and incorporates the intron sequence to the 5′side of the exon.	
256	c.600-124G>A	r.599_600ins600-216_600-128	p.Ile202LeufsTer62	Create a new donor site	Creates a donor site and an aberrant cassette exon in the intron.	
		r.599_600ins[(600-216_600-125;600-124g>a;600-123_600-1)]	p.Ile202LeufsTer37		Activates a cryptic acceptor site and incorporates an intron sequence to the 5′side of the exon.	
262	c.1717-1G>A	r.1716_1717ins[(1717-1897_1717-1679;1717-253_1717-2;g>a)]	p.Tyr573Ter	An acceptor site mutation	Activates a cryptic donor site and acceptor sites and creates a cassette exon and incorporates an intron sequence t theo 5′side of the exon.	
		r.1716_1717ins[1717-456_1717-2;g>a]	p.Tyr573SerfsTer110		Activates a cryptic acceptor site and incorporates the intron sequence to the 5′side of the exon.	
R19	c.1599+4A>C	r.1599_1560ins[gtgc;1599+5_1599+113]	p.Val534_Met535ins*10	A donor site mutation	Activates a cryptic donor site and incorporates the intron sequence to the 3′side of the exon.	

**Table 13 ijms-23-11175-t013:** Scoring system of clinical manifestations.

Score	Epilepsy	DD/ID	AF	HM	SEN	Cort. Tuber	Ret. Ham.	SEGA	Renal Cyst	Renal AML	Liver AML	Card. Rhabd.	LAM	MMPH	SP	UF	FP
**0**	**No symptoms**		**0~2 lesions**	**No lesion**					**No multiple lesion**			**No lesion**					
**1**	**Controlled**	**DQ/IQ = 50–69**	**Macular (flat) lesions only on cheek**	**1~2 lesions**	**Solitary lesion**				**All < 2 cm lesions**	**All < 1 cm lesions**		**Natural remission**	**Clinically no symptoms**				
**2**		**DQ/IQ = 35–49 ** **Or ** **Unknown severity**	**Papular lesions, < 3 mm diameter**	**3 or more lesions**	**2 to 5 lesions ** **Or ** **Multiple lesions of unknown exact number**		**Retinal achromic patch only**		**Max > 2 cm lesions**	**Max 1 to 4cm lesions**		**Positive lesions**					
**3**	**Active seizure**	**DQ/IQ < 34**	**Papular lesions, > 3 mm diameter, and/or extending to below mouth**		**More than 6 lesions**		**Multiple retinal hamartomas**	**positive lesion**	**Classic polycystic kidney disease**	**Max > 4 cm lesions**		**Congenital lesions detected by fetal USG**					

AF: facial angiofibroma; AML: angiomyolipomas; Card. Rhabd.: cardiac rhabdomyoma; Cort.Tuber: cortical tubers; DD/ID: developmental delay/ intellectual disability; FP: forehead plaque; HM: hypomelanotic macules; LAM: lymphangiomyomatosis; MMPH: Multifocal micronodular pneumocyte hyperplasia; Ret. Ham.: retinal hamartomas; SEGA: subependymal giant cell astrocytoma; SEN: subependymal nodules; SP: shagreen patch; UF: ungual fibromas.

## Data Availability

All *TSC1*/*TSC2* mutations described in this paper were registered in ClinVar (https://www.ncbi.nlm.nih.gov/clinvar/, last accessed on 15 August 2022). All genotype and phenotype data in this study are provided in Appendix A.

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
