# Peer review of "Genotype and Phenotype Landscape of 283 Japanese Patients with Tuberous Sclerosis Complex"

_ijms, 2022, doi:10.3390/ijms231911175_

Round 1

Reviewer 1 Report

This manuscript tasked to examine genotype-phenotype association of tuberous sclerosis complex (TSC) on Japanese patients.  Authors reported a total of 283 subjects with 225 definite, 53 possible, and 5 genetic diagnoses in 260 families (24 familial cases with 47 affected and 234 sporadic cases). Also reported was a total of 200 mutations including 64 TSC1 and 136 TSC2. Of the 283, 17 cases had mosaic mutations (2 in TSC1 and 15 in TSC2) and 11 had large gene deletion (2 in TSC1, 5 with TSC2 and 4 with TSC2/PKD1).  The study also reported 20% of subjects have mild features who could be missed from diagnosis (Dx). A higher frequency of TSC1 cases were observed with a low prevalence of epilepsy compared to TOSCA data. Authors suggested the higher utilization of brain imaging in Japan led to more Dx of mild neuropsychiatric cases. Authors also suggested global studies might not have correctly Dx TSC for the mild TSC cases.

Overall, the manuscript presented very valuable snapshots of the genotypes and phenotypes of a cohort of patients associated with TSC in Japan with good details. However, cautions should be taken to not to over generalize the study observations. Below are some major concerns on the study design and interpretations of the results:

Major concerns:

1.      This study cited the utilization of the 2021 published Pediatr Neurol “Updated International TSC Diagnosis Criteria and Surveillance and Management Recommendation”. However, the scoring system utilized in the study as shown in Table 12 appears to deviate from the criteria published in the cited Recommendations. For instance, manifestations shown in “Score 1” do not meet the Dx criteria shown in the cited Recommendation (e.g. solitary lesion of tuber, 1-2 lesions of hypomelanotic macule). Although epilepsy and DD/ID are common in TSC cases, they were not utilized as Dx criteria in TSC because these symptoms are not specific to TSC.  Comparing studies with different Dx criteria and stringency is questionable. 

2.       The study used two mutation detection methods (i.e. CHIPS and CoLAS) in two different subset of their subjects. The advantages of using CHIPS in detecting genetic variants are well recognized. However, the authors reported that CHIPS had a lower mutation detection rate than CoLAS. Question will be how many subjects with no mutation identified (NMI) in the CHIPS subset could have a mutation detected if they were examined by CoLAS? To this end, this study seems incomplete.

3.       Authors suggested the age group distribution in the study was similar to those of TOSCA data. However, Figure 1 in the manuscript clearly demonstrated this study had a much higher proportion of cases at or below the age of 5 years compared to TOSCA data (i.e. 46.3% vs 26.7% respectively). It is well recognized that many TSC phenotypes are age dependent, also described by the authors in the manuscript. It is not clear how the skew in this age group affected the analysis to conclude Japanese cases have milder phenotypes in general. It is not known why the authors want to compare their study with those in TOSCA knowing that TOSCA collected data from studies around the World except the America. 

4.       For any disease study, how the study subjects being recruited may lead to different observations and conclusions of the study. Authors should provide more detail descriptions on how their patients initially recruited/selected or referred to their study.  In addition, it is not known how to determine whether the subjects in the study can truly representing the overall TSC patients in Japan, as inferred by the authors in the manuscript. The data presented in the manuscript showed a high percentage of cases with cardiac rhabdomyomas including cases with regressed tumor (i.e. 50.5% vs 34.3% in TOSCA).

5.       It is not clear about how the authors obtained the “numbers of patients evaluated” that present in the “Tables” of the manuscript to perform statistical analysis. For example, Table 2 – showed “All age (225)” but Table_S1 in the Excel file showed 283 subjects with 3 “NE” subjects. Similarly, Table 2 showed “TSC2 (118)” and Table_S1 consisted 135 TSC2 subjects and 4 TSC2/PKD1 subjects. All numbers presented in the Tables of the manuscript should be verified to be consistent with the date sources in in Table_S1. Details in legend should be provided to help readers better understand the manuscript.

6.       In this study, “MMPH was found only in TSC1 patients” was concluded in the "Discussion" section.  However, there should be caution to point out most (209 of 283 subjects) were “NE” (i.e. not evaluated). Therefore there may be chances that some TSC2 patients may have MMPH like other published studies. Similarly, conclusions made about age-dependent phenotypes should also be presented with cautions.

7.   Similarly care should be taken to conclude that Japanese TSC cases have higher TSC1 mutations.  Most studies included only the probands of familial TSC cases in their genotype-phenotype association analysis. Table 5 appears to count all affected subjects with Definite TSC to obtain the ~32% TSC1 mutations to yield the conclusion that Japanese patients have significantly higher TSC1 mutations.  From Table_S1, we can find the TSC1:TSC2 mutation counts are almost the same (21:24) for familial cases and the Sporadic TSC cases have the TSC1:TSC2 counts at 43:116 (i.e. 27% were TSC1). Authors should make sure what is in the TOSCA numbers to make sure the comparison is apple to apple.

7.       Many contents in the Discussion appears to be results/observations. The manuscript can be more precise and/or concise.

Reviewer 2 Report

The manuscript entitled, "Genotype and Phenotype Landscape of 283 Japanese Patients  with Tuberous Sclerosis Complex " by Togi et al describes genotype and phenotypes of a large cohort of TSC patients. This outstanding descriptive manuscript provides relevant patient information including the presence of hamartomatous lesions, neurological presentation including epilepsy, and gene mutations. It would be worth providing additional information regarding sex and phenotypes, including LAM but also other lesions. The authors should also discuss whether any patients have both alleles mutated (not LOH within a lesion, but mutations outside of lesions) and discuss why patients do not have both alleles mutated.

Round 2

Reviewer 1 Report

Comment to A1.  - The added clarifications are good. Authors need to double check data because there remain cases that do not meet Definite Dx (e.g. families # 257, 36, 198, 99 and 128 with only one major feature with a score of 2). Also, a Possible TSC family #188 with score 2 on cortical tuber and hypomelanotic macules. These changes may lead to shifting of significance levels of the reported finding.  Also, usually only proband in familial TSC cases were utilized in genotype phenotype association studies. Utilizing both the proband and other affected members of familial cases in the association study will inflate the number of genotypes in the analysis.

Comments to A2.  It is unfortunate that around 40 families with NMI in the CHIPS subset have few remaining samples making it difficult to perform CoLAS.  The question remains how many of these NMI cases has a TSC1 or TSC2 disease causing variants and number may shift the significance of genotype phenotype associations. Authors may consider re-sample these cases to reach a more accurate association picture.

Comment to A3.  Points well taken. All studies had limitations and bias in some areas. While the combined TOSCA datasets appear to be least bias because the data came from large number of studies over the World other than the America, the study protocols varied in some ways between studies therefore shall be utilized with caution. The difference in percentages at and below 5 years is a good example.

Comment to A4.  Good to have the updated information about how the study cases were enrolled.  

Comment to A5.  Good to have the updated information. Again, usually only proband in familial TSC cases were utilized in genotype phenotype association studies. Utilizing both the proband and other affected members of familial cases in the association study will inflate the number of genotypes in the analysis.

Comment to A6.  Good to have the updated information. Clearly, the impact of genotype-phenotype association study relies on the completeness of genotype and phenotype information of each study case. More missing data reduce the impact of the study results.

Comment to A7.  Good to have the updated information.

Comment to A8.  Good to have the updated information.

Reviewer 2 Report

The revised manuscript is a significant contribution to the field and answers all questions that I have.

Thank you